# Widespread slowdown in short-term species turnover despite accelerating climate change

**Emmanuel C. Nwankwo** ⓘ **& Axel G. Rossberg** ⓘ ✉

When the species composition of ecological communities changes over time, environmental drivers are often invoked as the most plausible explanation. Several lines of reasoning, however, suggest that such compositional change, called temporal species turnover, can similarly result from intrinsic ecosystem dynamics, even in a constant environment. The degree to which these two drivers contribute to observed turnover remains unclear. To address this conundrum, we analyse the well-established BioTIME database of surveys. We expect either an acceleration of turnover with accelerating climate change or constant turnover if intrinsic mechanisms dominate. Surprisingly we find instead that species turnover over short time intervals (1-5 years) has decelerated in significantly more communities during the last 100 years than it has accelerated, typically by one third. The observed slowing of turnover, we argue, could be understood—when intrinsic dynamics dominate—as resulting because anthropogenic environmental degradation or declines of regional species pools reduce the number of potential colonisers driving turnover. Our results suggest that observed past changes in species composition were often manifestations of natural, intrinsic ecosystem dynamics. Although one can expect environmental drivers to dominate species turnover eventually as climate change accelerates further, for now such attribution should be done with caution.

Temporal species turnover, the ongoing replacement of the species forming an ecological community over time, is a widely observed ecological phenomenon[1]. The factors that can drive species turnover are varied[1–17]. In a decades-old debate, ecologists remain divided over the extent to which, amongst these, environmental changes or intrinsic processes are the dominating drivers[1–17]. Both sides can claim strong empirical support.

Changes in community composition are often conceptually decomposed into a turnover (replacement) component and additional species losses without gains or species gains without losses ('nestedness'). For clarity, we consider correspondingly any environmental change as decomposable in principle into, on one hand, change that benefits about as many species in the regional species pool as it is detrimental to and, on the other hand, change that is either detrimental or beneficial to all species, acknowledging that executing this decomposition in practice may be difficult. We will in the following call the first component *environmental shift* and the second *environmental degradation/amelioration* (depending on whether the change is detrimental or beneficial). By replacing species, environmental shift can directly cause species turnover, and faster environmental shifts will generally lead to faster turnover. Turnover (rather than nestedness) caused directly by environmental degradation/amelioration is more difficult to conceive.

Environmental shift, such as changes in temperature, precipitation or nutrient balance, can drive species turnover through direct effects on species and indirect effects, causing changes in species interactions[9]. Previous studies of the impact of environmental shift on turnover have, however, yielded inconclusive results[1–18]. For example, a

Centre for Biodiversity and Sustainability, School of Biological and Behavioral Sciences, Queen Mary University of London, London, UK.
✉e-mail: a.rossberg@qmul.ac.uk

recent meta-analysis of marine, freshwater and terrestrial community time series in Europe showed no significant effects of the rates of precipitation change on turnover rates and either faster or slower turnover with faster local warming, depending on site "naturalness", and yet evidence for acceleration of turnover over the last 20–40 years[6]. When characterizing community composition in terms of the average preferred temperature of species, clear temporal trends or correlations with local temperature trends can be found[14,15,19]. On its own, however, this is incomplete evidence, as an absence of change in average preferred temperature is not evidence for absence of species turnover. Despite these mixed results, it is widely agreed that sufficiently large environmental shifts will generate species turnover[2,3,18,20,21], potentially on a yearly time scale[6], but certainly over longer times as anthropogenetic climate change progresses.

Alternatively, species turnover can be driven by intrinsic processes, including ecological drift, evolution driven by species interactions, or a combination of species interactions and migration. The latter mechanism[16] operates as follows: if the regional species pool is large enough, some species from this pool will always be able to colonise a local community. Such colonisations then lead to disturbance of the local community, mediated by a complex interaction network (e.g., competition and predation), which can result in the extirpation of other species[16]. Often, the species being extirpated will have been locally rare and their extirpation be delayed[22]. Through this restructuring, the community then becomes open to colonisation by other species from the regional pool. The resulting turnover continues without end or direction in a kind of giant "rock-paper-scissors game"[23] played out on intrinsic population-dynamical time scales (typically years to decades)[16]. The celebrated Theory of Island Biogeography[24] and its later generalisation to spatially extended communities[11] predict similar non-directional turnover in community composition with apparently random colonisations and extirpations, although the theory does not specify the precise nature of the drivers[11]. Such ongoing non-directional turnover has since been documented on a decadal scale for island bird communities, with directional change arising only on a century time scale[5]. Intrinsically driven recurrent local turnover is also the mechanism thought to explain the 'shifting-mosaic steady state' spatiotemporal community pattern[12,25], a patchwork of different habitat types and their associated species that change over time and space[26,27].

To help resolve this debate about the causes of species turnover, our study exploits contrasting rates of global climate change that humanity has caused in a huge unintended experiment over the past 100 years. Interpreting a large collection of time series of community composition on this background we draw inferences about the ecological mechanism driving turnover.

Our study is mindful that the measured rate of species turnover and the dominant driving mechanism depend on the time lag between two observations of an ecological community[5]. Nowadays, turnover over lag periods of several decades is almost inevitably dominated by anthropogenic long-term environmental shifts. To capture potential intrinsic drivers, our approach, explained below, therefore focuses on short-term turnover with lag periods of just a few years. Yet, our approach can differentiate between environmental shifts and intrinsic processes as drivers of turnover. In particular, when short-term turnover is dominantly driven by directed environmental shifts, our method captures this equally sensitively as an analysis using longer lags.

## Results
### A robust methodology to measure turnover change
We developed an approach to study species turnover that is particularly sensitive to long-term changes in turnover rates and robust to variation in the quality of the available data[28]. It is based on the following considerations. Most studies of potential links between

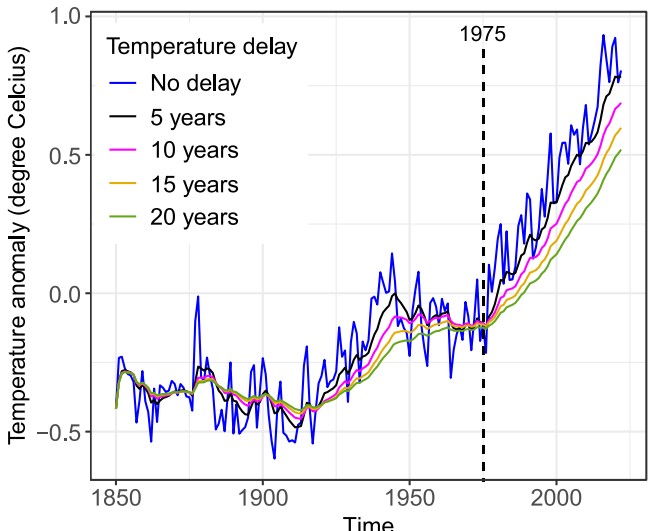

**Fig. 1 | Global surface atmospheric temperature (GSAT) time series.** We constructed delayed and smoothed GSAT time series, $GSAT_{del}$, using a simple recursive filter (see Discussion) with a delay parameter ranging from 5 to 20 years. Coincidentally, $GSAT_{del}$ change accelerates around 1975 for each delay-parameter value, potentially accelerating climate-driven species turnover.

turnover and environmental shift focus on local climatic patterns as the main drivers[2,6,14]. The implied assumption, which we follow here, is that such local climate change contributes substantially to the local environmental shifts potentially driving species turnover. Long-term changes in local climatic patterns (temperature, precipitation, cloud cover, wind, etc.) are, in turn, driven by changes in Earth's atmospheric energy balance, which is quantified by changes in global mean surface air temperature (GSAT)[29]. Hence, the well-documented accelerating GSAT increase since around 1975 (Fig. 1) was generally accompanied by accelerating local environmental shifts, also where records of these local shifts are incomplete[30]. For example, IPCC's WG1 Fifth Assessment Report[30] (their Fig. SPM.2) demonstrates: (i) an acceleration of precipitation change (absolute rates of change tend to be higher when averaged since 1951 than when averaged since 1901), (ii) spatial variation in the direction of change (documenting positive and negative change), but also (iii) large gaps in spatially resolved global data on this acceleration. The GSAT time series in Fig. 1 captures the driver of such incompletely documented changes in local climates: changes in the global atmospheric energy balance.

To assess the impact of these local environmental shifts on species turnover, we analysed the BioTIME database[28] of timeseries of community composition. We compared turnover rates in BioTIME community timeseries for the periods up to and since a given breakpoint year hypothesised to mark the acceleration of environmental shifts[31], and tested for a deviation from random expectation of the numbers of communities for which turnover accelerated and decelerated. This approach avoids the need to first identify and document at each location the climate variable most likely affecting turnover and then correlating this variable with turnover[2,6], thus avoiding potentially substantial loss in statistical power.

The precise year in which the GSAT increase accelerated is difficult to pinpoint. Moreover, one would expect that species turnover would respond with some delay to this acceleration. Rather than comparing turnover rates before and since just a single breakpoint year, we therefore repeated our analysis for a wide range of conceivable breakpoint years that covers the period of interest in the 1970s and 1980s. If in this period of accelerating environmental change (Fig. 1) the expected acceleration of species turnover indeed occurred, we'd expect that for at least one of the breakpoint

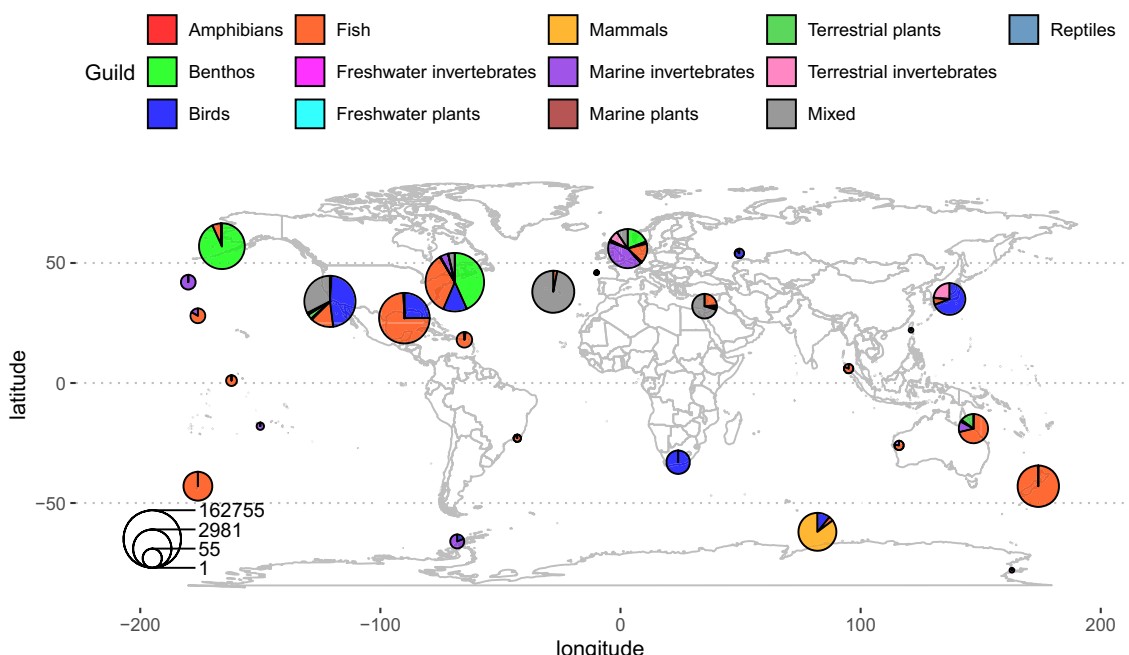

**Fig. 2 | The distribution and types of communities used in this study after filtering.** The size of each pie indicates the number of records of species observations, which controls statistical power. To simplify the graphic, communities are grouped using a leader clustering algorithm[70] with 2000 km radius. Map outlines generated using Natural Earth data.

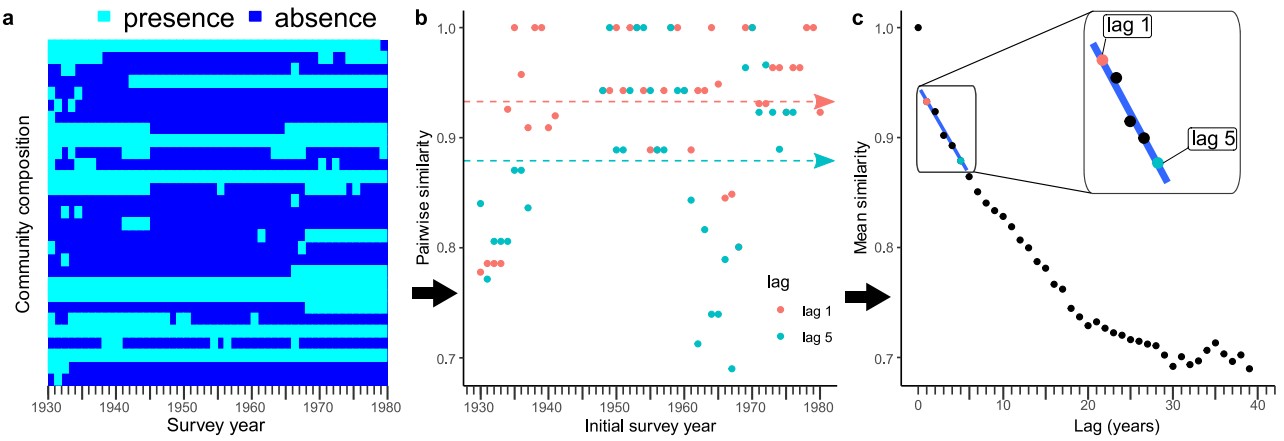

**Fig. 3 | Estimation of turnover rates. a** The temporal changes in community composition for a temperate bird community[71] (BioTIME Study ID: 46); **b** Community similarity for surveys separated by a lag of one year (lag 1) and five years turnover since this year is faster than before in most communities.

years (lag 5), with time-averaged similarities indicated by dashed arrows; **c** Turnover rate is computed as the regression slope of the decline of time-averaged similarity with lag over 1–5-year lags.

years turnover since this year is faster than before in most communities.

The BioTIME database includes community composition datasets with considerable temporal extent (up to 44 years, median 7 years) and spatial coverage[32,33] (Fig. 2). To minimise the risk of confirmation bias, which would arise if we'd preferentially include studies in our analysis that fit our preferred hypothesis, we applied our analysis directly to this public database, without adding and with minimal filtering of studies (Methods). We partitioned studies covering more than 96 km² into separate local communities, following a previously described spatial data partitioning protocol[2,17], again to avoid confirmation bias. The spatial distribution and type of the partitioned communities is illustrated in Fig. 2.

We compared communities through time based on species presence/absence data (Fig. 3a) to avoid a loss of statistical power associated with domination of metric values by a few abundant species. Community similarity was quantified using the Ochiai metric[34], which is known to have low sensitivity to the presence of rare species and be robust to variation in sampling effort[35,36].

To guard against pseudo turnover[37–40], which arises when species that are present at a site are overlooked in some of the yearly surveys but recorded in others, we decided not to quantify turnover directly in terms of the similarity of communities from one survey year to the next[6,41–43], which would correspond to a 1 year lag in Fig. 3c. Instead, we measured turnover as the rate at which average similarity declines with increasing lag between pairs of surveys separated by small lags. We verified in model computations that this choice does not affect our ability to quantify directed turnover driven by environmental shifts when this driver dominates over short lag periods (Supplementary Code 1).

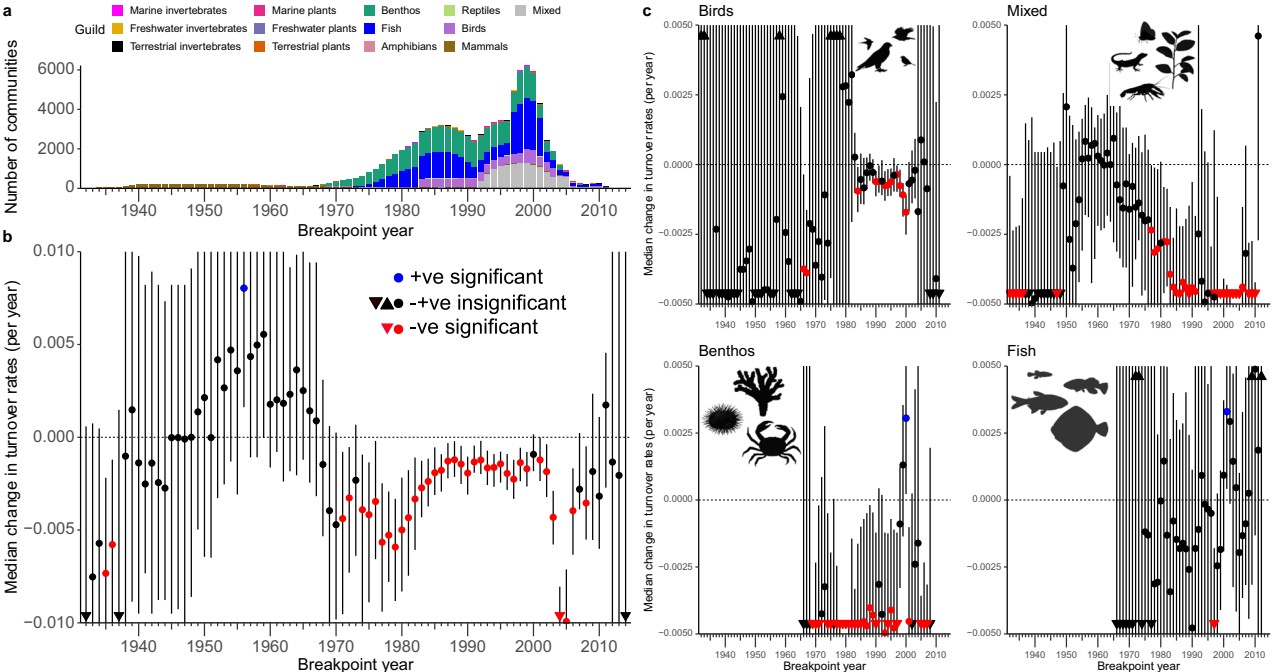

**Fig. 4 | Deceleration of turnover across community types. a** For each breakpoint year, the number and type of communities included in the analysis; **b** Median change in turnover rate for each breakpoint year (circles, triangles for outliers) with 95% confidence intervals (bars). The change was negative, indicating that turnover before the breakpoint year was faster than since, for almost all breakpoint years with statistically significant change. **c** Separate analyses for the main community types entering panel (**b**). Each of these four types is represented by over 200 communities in some breakpoint years (cf. panel **a**). N.B.: dependence of results on breakpoint year cannot be interpreted as an accelerating or decelerating change in turnover rate because the set of studies evaluated depends on the breakpoint year, amongst other reasons. The samples sizes corresponding to the confidence intervals in panels (**b**) and (**c**), i.e., the numbers of biological replicates (communities) analysed, vary by breakpoint year and are shown in panel (**a**). For panel (**b**), sample sizes and p-values for each breakyear are also listed in Supplementary Table S1. Illustrations created in BioRender, Rossberg, A. G. (2026), https://BioRender.com/qsc34tn, https://BioRender.com/nx85f4d, https://BioRender.com/xrj7mmr, https://BioRender.com/4alakst.

Specifically, to determine for a given community its turnover rate for the period *before* a chosen breakpoint year: (1) We calculated, for a fixed lag $l$ (varied from $l = 1$–5 years), the average Ochiai similarity for all pairs of surveys separated by $l$ years that took place before the breakpoint year (as conceptualized in Fig. 3b for one such period). (2) We then determined the turnover rate for this pre-breakpoint period as the rate at which average similarity declined with lag[5,44,45], regressing over lags 1–5 (Fig. 3c). After repeating this analysis for the available surveys of this community since the breakyear, we computed the difference between the turnover rates since and before the breakpoint, such that an acceleration of turnover corresponds to a positive and a deceleration to a negative difference. Finally, we computed the median of this difference over all communities and the corresponding 'exact' confidence interval[46]. We used the median rather than the mean to improve robustness against outlier studies.

This analysis doubles up as a statistical significance test for deviations from the simple null hypothesis that accelerating and decelerating turnover is observed with equal probability: a confidence interval for the median that lies entirely above zero implies that the number of communities exhibiting accelerating turnover is significantly larger than the number exhibiting decelerating turnover and vice versa (two-sided test at 95% confidence level).

### Decelerating turnover

Much to our surprise, we found that for nearly all choices of breakpoint year where our analysis could demonstrate a statistically significant change in median turnover rate at all, this change was negative. That is, despite accelerating climate change (Fig. 1), species turnover tended to decelerate (Fig. 4b). The only case of significantly positive change, the 1956 breakpoint (Fig. 4b), disappeared when we repeated the analyses

using different community similarity metrics or lags longer than 5 years to compute turnover rates (Supplementary Figs. S1, S2). For those cases where positive change was statistically excluded, median change values ranged from −0.01143 $yr^{-1}$ to −0.00121 $yr^{-1}$ (median over all breakpoint years: −0.00272 $yr^{-1}$), which is considerable compared to the corresponding median overall turnover rates for the same set of breakpoint years, which ranged from 0.00469 $yr^{-1}$ to 0.01239 $yr^{-1}$ (median: 0.00627 $yr^{-1}$).

Sample sizes (i.e., the number of separate community time series contributing to medians) varied substantially over breakpoint years, ranging from 14 to 6051 (mean of 1170, Fig. 4a). For early breakpoints (period 1932–1970) and very late breakpoints most confidence intervals span zero in Fig. 4b, implying that the analysis remains inconclusive, which is explained by the small amount of data available to estimate turnover rate either before or since these breakpoints (Supplementary Table S1).

### Verification of robustness and generality
Testing the robustness of the analysis to the use of other metrics of community similarity (Supplementary Fig. S1) or consideration of community change over lags longer than five years (Supplementary Fig. S2), we found similar patterns in the overall results.

As further tests for robustness, we repeated the analysis after removing from the published data partitioning protocol[2,17] a step filtering for 'sample completeness'. Hypothesizing that the observed changes in turnover rate might have resulted from intentional manipulations of study communities, we repeated the analysis also after removing all studies for which BioTIME registers a 'treatment'. Results remained essentially unchanged in both cases (Supplementary Figs. S3, S4).

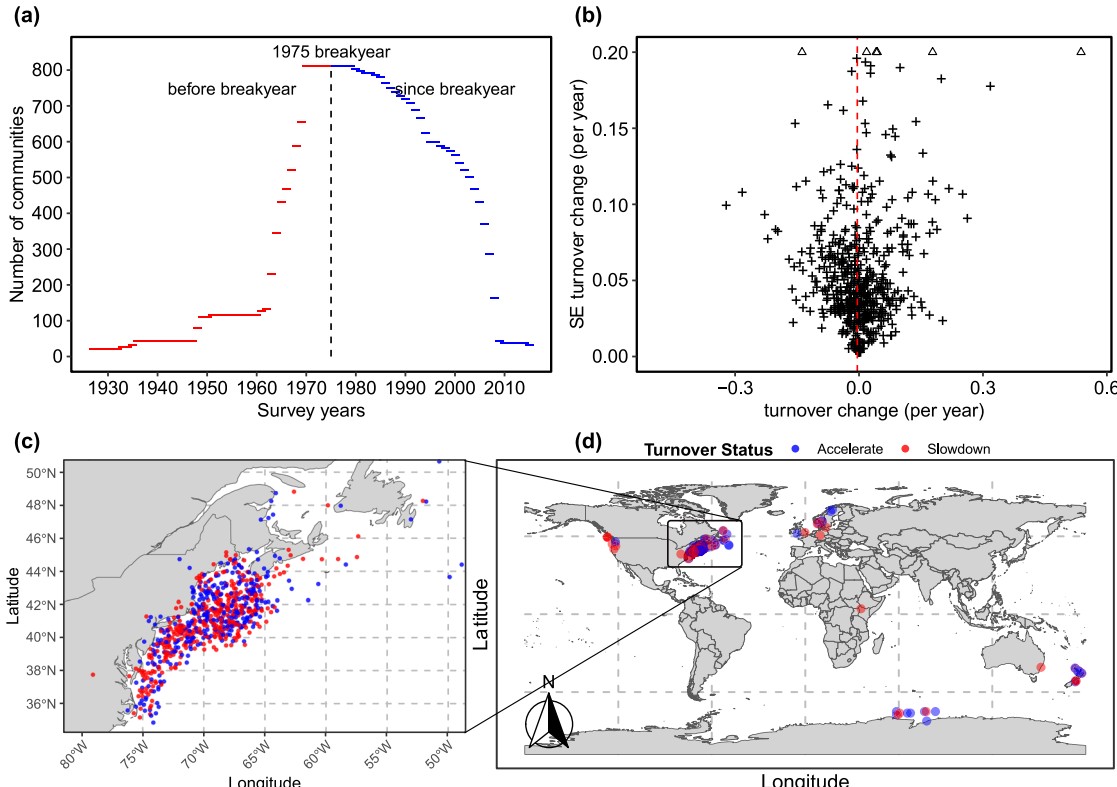

**Fig. 5 | Data underlying the 1975 breakpoint year. a** Coverage of periods before and since the breakyear by the community time series entering the 1975 beakyear analysis. **b** Shows that, with decreasing standard error (SE), estimated turnover change for the $n = 809$ distinct communities analysed comes closer to the community median (vertical dashed line), usually within 2 SE. The median is significantly below zero, but variable data quality requires careful analysis to reveal this. **c**, **d** show that there is no obvious associate of positive or negative turnover change with study location. Source data are provided as a Source Data file. Map outlines generated using Natural Earth data.

To test the generality of our results across community types, we repeated the analysis separately for the four types that contribute most to the overall result (Fig. 4a): benthos, fish, birds, and the mixed communities labelled 'All' in BioTIME[28]. We found the observed slow-down replicated independently for mixed, bird, and benthic communities (Fig. 4c). For fish communities, there was no consistent signal (Fig. 4c), presumably reflecting that most underlying data are surveys of exploited fish communities where fisheries management disrupts natural community dynamics.

## Discussion

Ecological community composition generally responds to environmental shifts not instantaneously but with some delay[3] if at all. This makes it difficult to pinpoint a particular year in the GSAT time series (Fig. 1) as an adequate breakyear for our analysis. For illustrative purposes, we constructed delayed GSAT time series, $GSAT_{del}$—quantifying hypothetical delayed community responses—using the simple recursive filter $GSAT_{del,y} = \exp(-1/d)\, GSAT_{del,y-1} + [1-\exp(-1/d)]\, GSAT_y$, with a delay parameter $d$. This filter is a simple model for delays arising from natural mechanisms, which are intrinsically accompanied by a smoothing of the delayed responses, as illustrated in Fig. 1. Here, this smoothing removes much of the short-term fluctuations of GSAT, exposing long-term trends. By a fluke of chance, these trends exhibit for any $d$ in the range 5–20 years[3,47] a pronounced acceleration in 1975 (Fig. 1). To explain the structure of our argument, let us therefore assume that 1975 is an adequate breakyear, such that local environmental shifts, taking delay and smoothing of community responses into account, were generally slower before this year than since. If short-term turnover was dominantly driven by environmental shifts, it should thus generally have been slower before 1975 than since.

But for the data in BioTIME this is not the case. The median difference between the turnover rates since and before 1975 for all community time series that straddle 1975 sufficiently to permit this analysis is significantly negative, implying that turnover has become slower in most cases (Figs. 3a and 4b). Certainly, it has not generally become faster as one would expect for turnover driven by environmental shifts. We find a 33% deceleration in median turnover rate since the breakpoint compared to before (median turnover rate before 1975 breakyear = 0.00812 yr⁻¹, since 1975 breakyear = 0.00547 yr⁻¹). This analysis includes data from 23 studies from marine, freshwater, and terrestrial habitats, including birds, mammals, fish, plants, and invertebrates (Fig. 5c, d, Supplementary Table S3).

The result is not due to a mismatch between the time periods covered by the community time series and the period in $GSAT_{del}$ over which the acceleration occurred. Barely any community data set entering the analysis reaches back into the 1940s (Fig. 5a) where $GSAT_{del}$ becomes more complicated (Fig. 1). While there is also a good proportion of community time series for which the difference in turnover rates we obtain is positive, our analysis in Fig. 5b suggests that this may in many cases be the result of high measurement uncertainty. The standard error of the difference in turnover rates is for most community time series large compared to the median change that we find (Fig. 5b). It is only by pooling the BioTIME data that the tendency of species turnover to become slower can be demonstrated.

Crucially, the general conclusion from this argument does not depend on an assumption that 1975 is the relevant breakpoint year. For almost all breakpoint years between 1971 and 2006 (with exceptions

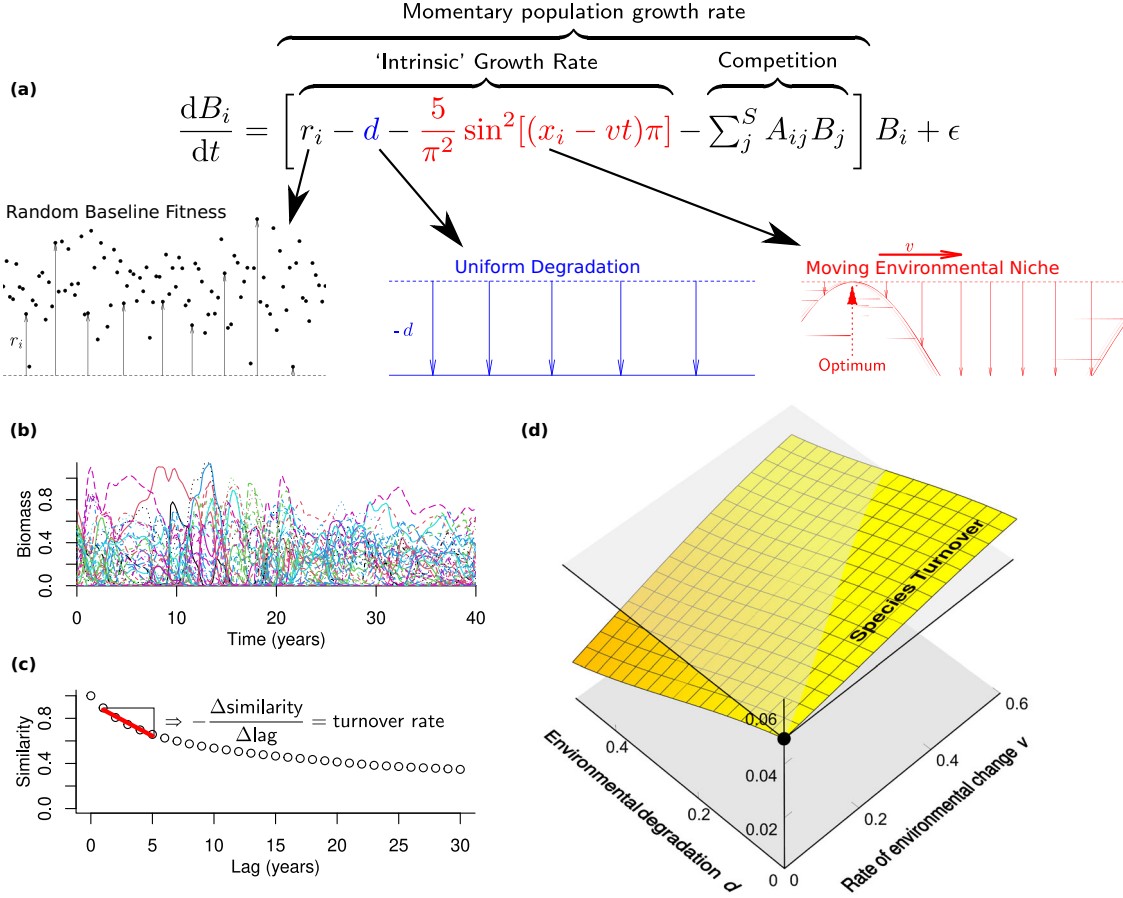

**Fig. 6 | Model explaining slowdown.** We developed a numerical community model (**a**) that explains the unexpected decline in turnover rate (**b**, **c**) in the face of accelerating environmental changes. Our model takes into account the combined effects of environmental change and intrinsic dynamics, as well as the population dynamical effect of a hypothesized overall constant environmental degradation. To represent this degradation, we reduced the population growth rate in the absence of competition for all species by the same amount. From simulated community timeseries as shown in (**b**), with species differentiated by colour and line style, we compute the turnover rate applying the same method as used for empirical data (**c**). Our model shows that turnover slows down as the environment degrades and increases with faster environmental change (**d**). The transparent horizontal plane in (**d**) represents the turnover rate without environmental change or degradation for comparison. Equation (**a**) describes a Lotka-Volterra competition model for

S = 1470 species. $B_i$ stands for the biomass of species $i = 1, ..., S$; $r_i$ for its intrinsic growth rate under optimal conditions, sampled randomly with mean 1 and standard deviation 0.25; $d$ (varied between 0 and 0.5) represents the constant environmental degradation; and the term involving $\sin^2$ the environmental preferences of species, with species' preferred niches $x_i$ spaced evenly between zero and one. We modelled niches using a sin function to avoid boundary effects at $x = 0, 1$ that would complicate interpretation of results. The environment changes with a rate $v$ (varied from 0 to 0.6). The $A_{ij}$ represent competition coefficients and $\varepsilon = 10^{-20}$ a weak rain of propagules that permits species to re-invade after their effective local extinction. Following ref. 22, we randomly set the competition coefficients $A_{ij}$ to 0.4 with probability 0.4 or otherwise set them to 0, except for the intraspecific competition coefficients $A_{ii}$, which we all set to 1 for simplicity.

for 1973 and 2000) we find a statistically significant deceleration in turnover. For most other years the data is insufficient to determine a statistically significant median change (Fig. 4b). From these observations we conclude that, at least for the types of communities covered by BioTIME, environmental shifts have not been the dominant driver of species turnover over short lags in the past century. This would be inconsistent with the deceleration of turnover that we have demonstrated.

### Explaining the decline of turnover rates

To further probe the plausibility of this conclusion, we shall now discuss whether under the alternative hypothesis, which is that short-term turnover is dominantly driven by intrinsic processes, the observed deceleration of turnover could be explained.

For this, we implemented a simple numerical community model known to exhibit intrinsically driven turnover, as explained above, by

remaining open to colonisation from a large species pool[16,48,49], and extended this model to account for the combined effects of environmental shift and a hypothesised overall environmental degradation (Fig. 6). We modelled environmental shift by changing the value of an environmental variable at a fixed rate—given by a model parameter—relative to the preferred niche values of species. Degradation was modelled by a simple overall reduction of the intrinsic growth rates of species (i.e. their growth rates in the absence of competition) by an amount given by a second model parameter. In reality, this degradation could correspond to a combination of different anthropogenic pressures including habitat destruction or environmental pollution.

Compared to a baseline scenario without environmental shift or degradation, we find that accelerated environmental shift accelerates species turnover in the model, while a degraded environment leads to slower turnover on average. The two effects

balance each other for certain combinations of degradation and shift rate, such that the turnover rate does not change. The slower turnover in degraded environments can intuitively be understood from the fact that the modelled site becomes suitable for fewer species when degraded, which reduces the number of potential colonisers driving turnover.

The model simulations confirm that if environmental shift was the dominant driver of species turnover our findings analysing the Bio-TIME database[28] would be hard to explain. On the other hand, the simulations demonstrate that the observed deceleration of turnover could be understood if, on a background of intrinsic turnover dynamics, the net impact of various forms of environmental degradation was slowing turnover less before the given breakyear than since that breakyear, overriding potential acceleration caused by accelerating environmental shifts. This degradation might be due to land-use change[50] or environmental pollution arising from the production, use and disposal of pesticides, pharmaceuticals and heavy metals, which have been implicated in impacting reproductive physiology, sexual communication, sexual selection and parental care[20] and have ecological consequences for the populations exposed to them[51]. Such an attribution of decelerating turnover to the intensity of environmental degradation is consistent with a study of turnover in bird communities in a tropical Andes conservation hot spot[52] that concluded: "It is striking that the less altered habitat, native forest, has a higher rate of composition change than the more altered shrub and introduced forest habitats".

Other kinds of environmental pressures could contribute to a deceleration of environmental change as well. For example, species turnover can be expected to decelerate when the regional pool of potential coloniser species declines[1] or when habitat fragmentation limits the dispersal of propagules of such species[53,54]. We illustrate the pool-size effect for the model of Fig. 6 in Supplementary Fig. S5. In a more detailed spatially extended variant of this model both the decelerating effects of declining species pool and habitat fragmentation have been demonstrated[16]. Overall, the specific causes of the slowing of species turnover may be complex and multifaceted. We therefore suggest further investigation of this phenomenon. For example, consistent with the interpretation above, our measure of turnover was for North American bird communities recently shown to systematically decline with increasing human habitat modification, while it increased with increasing regional species pool[45].

There may also be other, yet-to-be-identified explanations that could account for the slowing of species turnover we observe. These include factors such as improved environmental management and changes in sampling methodology, which could affect the accuracy of data collected on species turnover over time.

We also considered the possibility that undirected short-term weather and climate variability might dominantly be driving species turnover over short lag periods. However, these variations are becoming more intense as global warming progresses[55,56] and therefore could not easily explain the slowing of turnover that we documented. This includes the El Niño-Southern Oscillation phenomenon, the main cause of the variability[57] seen in the GSAT time series in Fig. 1 since 1950 (before 1950 measurement uncertainty dominates), which has intensified[58] since ca. 1960.

### Reconciling dominance of intrinsic drivers with observed climatic effects

Our study suggests that in the past century climate-driven species turnover may not have been as strong as previously thought. However, this does not mean that climatic effects don't exist. As Fig. 6 demonstrates, intrinsic and environmental drivers of turnover seamlessly combine. In the future and over longer lag periods, one can expect the effect of environmental shifts on species turnover to become stronger[16]. Indeed, directional trends in shifts of species ranges consistent with climatic shifts are now well documented[59–61] and these will inevitably contribute to local species turnover[62]. The fact that these trends emerge from a background of apparent idiosyncrasy in range shifts, where ranges often shift in a direction opposite to what is expected from climate change, is consistent with our conclusion that changes in local species composition are driven to a large part by intrinsic dynamics that cause apparently idiosyncratic colonisations and extirpations.

Consistent with the expectation that the relative strength of climatic effects will gradually increase is also an analysis of European community time series focussing on the turn of century and early 21st century, i.e., a period later than the bulk of BioTIME data, which reported significant acceleration of turnover for terrestrial systems (but a deceleration for some marine systems)[6]. Furthermore, the expectation that over longer time scales the relative strength of environmental drivers of turnover becomes stronger[45] is consistent with a study of the BioTIME database that quantified turnover based on community similarity over lags up to the full length of empirical time series (rather than just up to five years), which reported significant correlations between turnover rates and absolute rates of temperature change[63].

Overall, our study highlights the importance of understanding the drivers of species turnover for the interpretation of biodiversity dynamics on a changing planet. Species turnover can be a natural phenomenon indicative of a healthy environment, a large regional species pool, and good connectivity between patches with similar habitats. Changes in turnover rates can also be the result of disruption of communities by directed environmental shifts, although our study suggests that this was not the dominant driver in the past century. By identifying the underlying causes, managers and regulators can develop more effective strategies for assessing and protecting our natural environment.

## Methods
### Selection of similarity metric

Our method to determine species turnover rate is applicable to a wide range of conceivable indices for the compositional similarity of communities. Independent of similarity index, we found that typically, as illustrated in Fig. 3c, the decline in similarity from lag 0 (where similarity is 1 by construction) to lag 1 was much larger than the mean rate of decline of similarity with lag for non-zero lags. This indicates high pseudo-turnover due to false negative observations, which not only biases similarity indices but can also generate random variability (noise).

As a first, coarse selection criterion, we therefore determined for each community and for each of ten similarity indices enumerated in ref. 64 a 'signal to noise ratio' as the ratio between the difference of the similarities for lags 1 and 5 (as the 'signal') and the difference of the similarity for lags 0 and 1 (which is dominated by false negatives, the 'noise')---discarding communities where data was insufficient to compute this ratio. The median of this 'signal to noise ratio' was highest for the Sørensen (0.087) and the Ochiai (0.088) similarity indices.

To accurately differentiate between Sørensen and Ochiai index, we derived for both indices approximate analytic formulas for the coefficient of variation (CV) of the difference between similarity values computed for two different non-zero lags in the presence of random false negatives. We then showed using computer algebra that for false negative rates smaller than 2/3 and for a reasonably slow decline of similarity with lag this CV is, all else equal, always smaller for the Ochiai index than for the Sørensen index (see Supplementary Note 1 for details). We concluded that in the context of our study the Ochiai index is more robust to false negatives than the Sørensen index and therefore used the Ochiai index in our main analyses.

### Estimation of turnover rates

The formula for the Ochiai index of community similarity is:

$$\text{Ochiai index} = \frac{a}{\sqrt{(a+b)(a+c)}} \qquad (1)$$

where $a$ is the number of species shared by the two communities, $b$ the number of species unique to the first community, and $c$ the number of species unique to the second community. The Ochiai index has been shown to perform well across a range of community types and ecosystems, including both plant and animal communities, marine and freshwater habitats, and terrestrial ecosystems[35,36,65]. Because species richness displays only weak trends at community level in BioTIME (Supplementary Fig. S7 of ref. 17, taking account the artifacts in data before 1927 we observed), we did not attempt to separate so-called nestedness and species-replacement components in the Ochiai index[66]; in a previous study using a similar method, such separation had little effect on results[45].

We noticed that some communities surveyed earlier than 1927 displayed large differences amongst early years, which may be an artefact of the early stage of evolving ecological survey techniques, skills, and interests. Hence, data from survey dates in each time series earlier than 1927 were filtered out (38,003 observations). Overall, after all filtering, we analysed 4,863,824 records of species observations from 189 studies (listed in Supplementary Table S2) comprising 15,469 species and seven guilds (plants, fish, birds, mammals, reptiles, invertebrates, benthic and mixed communities) from freshwater, marine, and terrestrial ecosystems (Figs. 2 and 4a).

For each breakyear, we included all those communities in the analysis whose time series started more than 5 years before and ended at least 5 years after the breakyear. When using longer lags to compute turnover these periods were extended correspondingly, leading to more exclusions and so smaller samples sizes for medians over communities.

Some communities did not have records for each year between start and end of the timeseries. For each community and each lag value, we averaged similarity only over those year-pairs for which data was available. Correspondingly, we computed a community's turnover rates from the regression of mean similarity against lag using only those lag values for which at least one year-pair was available. Communities with less than two data points available for this regression were excluded.

For some communities, the number of samples (e.g., hauls) taken varied amongst survey years. Do avoid a potential bias of similarity values by this variation, we first determined the minimum number of samples taken in a survey year and then computed rarefied similarities between all survey year pairs by averaging similarities computed from 100 sub-samples of the community time series, obtained by randomly retaining for each survey year only this minimum number of samples. Mean similarities by lag (Fig. 3c) were then computed from these rarefied similarities for the periods before and since the given breakpoint years.

By computing species turnover through a regression of similarity against lag (Fig. 3c)[45], we circumvent a potential bias by pseudo-turnover because this bias can be expected to be the same for all lags larger than zero[44].

To estimate standard errors of turnover change in Fig. 5b, we first estimated for each community and break year the standard errors for turnover in the periods before and since the break year by the nominal standard error of the linear regressions in Fig. 3c (excluding cases where data for only two lag values was available). Then, following standard propagation-of-errors rules, we estimated the standard error for turnover change as the square root of the sum of the squares of these two estimated standard errors.

### Communities with insufficient data

For some communities, the differences of the turnover rates we obtained before and since a given breakpoint year was exactly zero. In most of these cases, both turnover rates were zero. Generally, these communities were represented by time series recording just a few species that may have changed in their abundances but remained present throughout.

For the following two reasons, we decided to discard these community/breakpoint combinations when computing median changes in turnover rates and corresponding turnover rates. Firstly, it is likely that surveys of these communities with a wider scope of target species would have revealed some turnover and changes in turnover rate, and if with some measurement error. Inclusion of the zeros might therefore bias estimated median differences of turnover rates. Secondly, while there is the obvious null hypothesis of equiprobability of sigs when all differences of turnover rates are either positive or negative, hypothesis testing accounting for the possibility of zeros is more ambiguous---and yet might boil down to comparing the numbers of positive and negative values only[67].

In spirit, this approach is similar to the step of filtering out community time series reporting only small numbers of species implemented in related work[63].

### Robustness analysis

To test the robustness of our findings, we repeated our analyses using other metrics of community similarity (e.g. Jaccard and Sørensen) implemented within the ade4 R package[68] and examined community change over lags $l$ longer than five years.

### Simulation of changes in turnover rate in response to environmental change and degradation

We developed a numerical model (Fig. 6a) to elucidate the unexpected decline observed in the turnover rate (Fig. 6b, c). The simulation was conducted over 4000 unit times with the sampling interval ('year') given by 25 unit times. A burn-in period of 400 unit times was discarded to allow the system to reach a steady state before data collection. To determine the species turnover rate for each scenario of environmental change and environmental degradation, we utilized the method described for empirical data analysis, classifying all those species $i$ as present whose simulated biomass $B_i$ exceeded 0.01. We simulated this model in a full-factorial design with 20-fold replication, varying the intensity of environmental degradation $d$ from 0 to 0.5 in 20 steps and the environmental rate of change $v$ from 0 to 0.6 in 20 steps (totalling 8000 model runs). Panel (d) in Fig. 6 shows a full third-order multivariate polynomial fit to the dependence of turnover rate on $d$ and $v$. The Akaike Information Criterion preferred this third-order fit over second- or fourth-order fits.

As a strategy, the model parameters in Fig. 6 were chosen for easy reproducibility by keeping the number of modelled species and the duration of the simulations small but large enough to generate effects emerging at large local and regional species richness. The distribution of intrinsic growth rates was chosen to obtain a wide distribution of values but only few negatives. The chosen width of the environmental niche can select several non-overlapping sets of species as the environment shifts. With the chosen parameterization of the competition matrix $A_{ij}$, the theoretical species richness at which ecological structural instability[16] arises in this model, $S_{ESI}$ in ref. 69, is 18.375, just sufficient for high-richness phenomena to emerge. The size $S = 1470$ of the simulated species pool was chosen as $S = 80\,S_{ESI}$ to obtain a sufficiently large number of environmentally viable species to yield intrinsic local turnover despite the limited width of the moving environmental niche in the model. We repeated the simulations with $S = 40\,S_{ESI}$ and $S = 120\,S_{ESI}$ without observing a qualitative change in outcomes (Supplementary Fig. S5).

**Reporting summary**

Further information on research design is available in the Nature Portfolio Reporting Summary linked to this article.

## Data availability

The data underlying this study were sourced from the BioTIME database[28]. A file listing for each breakyear, lag, metric and community the corresponding turnover rates before and since the breakyear, has been deposited in the Zenodo database with [https://doi.org/10.5281/zenodo.17791135]. Source data are provided with this paper.

## Code availability

The code used for our analyses has been deposited in the Zenodo database with [https://doi.org/10.5281/zenodo.17791135].

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

## Acknowledgements

The authors acknowledge discussion of this study with Chris Terry. This work was supported by the Natural Environment Research Council (NERC) [grant number NE/T003510/1] (AGR).

## Author contributions

A.G.R. acquired funding, and designed and conceptualised the study. E.C.N. developed and tested the methods, performed the data analyses, and wrote the original draft. E.C.N. and A.G.R. jointly tested the code, reviewed, and edited the manuscript and developed data visualisation.

## Competing interests

The authors declare no competing interests.
