## [Transparent Peer Review file · Nature Communications]

Widespread slowdown in short-term species turnover despite accelerating climate change

Corresponding Author: Professor Axel Rossberg

Version 1:

Reviewer comments:

Reviewer #3

(Remarks to the Author)

My sincere thanks to the authors for the care, rigour and open-mindedness with which they have addressed my concerns regarding this manuscript. As a result of their detailed response to the various issues I raised, I am now of the opinion that the analysis they have undertaken is indeed scientifically sound. However, I do feel that the framing of this analysis, particularly in terms of its purpose and underlying assumptions, needs to be conveyed even more explicitly and unambiguously than it is at present.

In their response to my concerns – particularly under “Comment 2” and “Comment 3” – the authors have further clarified that the “species turnover” they are focusing on here is solely that operating at a time scale of no more than 5 years, i.e. “short-term turnover”. If one accepts this particular focus, as I have now done, then most of the other issues I raised in my initial review become of far less concern. However, I do remain concerned that the nature of, and reason for, this particular focus on short-term turnover rates rather than on longer-term trends of change in community composition is likely to be lost on many readers of this manuscript in its present form.

Even the paper’s title – “Widespread slowdown in species turnover despite accelerating climate change” – risks giving many readers (including me on my first reading) the impression that this study will be addressing longer-term directional changes (i.e. trends) in community composition, rather than shorter-term fluctuations around such trends – as the former is the time scale at which a large body of previous literature has suggested impacts of climate change on biodiversity are actually playing out. I feel that the risk of readers assuming the current study is, or should be, focusing on longer-term trends in the impact of climate change is then not helped by the way the authors introduce and explain the central role played by the “accelerating GSAT increase since around 1975” in their analysis. As clearly depicted in Fig 1 of the manuscript, this acceleration is in the longer-term rate (trend) of warming rather than in the year-to-year fluctuations around this trend. Given the emphasis placed by the authors on this before-versus-after-1975 acceleration, less astute readers could well be forgiven for assuming that the study’s analysis of rates of species turnover is, or should be, also focused on longer-term trends in community composition.

I suggest the following to help reduce the overall risk of readers misinterpreting the purpose, nature and conclusions of the analysis undertaken in this study:

1. Modify the paper’s title to clarify the short time scale over which species turnover is being addressed here – e.g. “Widespread slowdown in short-term species turnover despite accelerating climate change”.
2. Early in the Introduction explain, even more clearly, the particular time scale of interest in this study’s assessment of rates of “species turnover”, and the reason why the authors chose this focus, as opposed to focusing on longer-term (directional) trends in compositional turnover under climate change.
3. In the “A new methodology” subsection of the Results, also explain more clearly the relevance of the acceleration in GSAT increase after 1975 to the presented analysis of rates of species turnover before and after this time point given that

(judging from Fig1) the former is an acceleration in a trend playing out over a much longer time scale than the latter. Given the 5-year time window over which species turnover was assessed appears to correspond more to the year-to-year fluctuations around the GSAT trend depicted in Fig 1, rather than the trend itself, a more astute reader is likely to wonder why the authors are placing so much emphasis on the change in long-term GSAT trend rather than the change (if any) in variability around this trend before-versus-after 1975. This issue deserves more attention.

(Remarks on code availability)

Responses to reviewer comments on revised manuscript NCOMMS-25-66173A “Widespread slowdown in species turnover despite accelerating climate change”

We thank Reviewer 3 for their positive assessment of our manuscript and our responses to their previous comments. Reviewer 3 made three specific recommendations to make purpose, nature and conclusions of our analysis clearer to our readers. Below we explain how we took up each of these recommendations. Reviewer recommendations are in blue, our responses in black.

1. Modify the paper’s title to clarify the short time scale over which species turnover is being addressed here – e.g. “Widespread slowdown in short-term species turnover despite accelerating climate change”.

We followed this suggestion.

2. Early in the Introduction explain, even more clearly, the particular time scale of interest in this study’s assessment of rates of “species turnover”, and the reason why the authors chose this focus, as opposed to focusing on longer-term (directional) trends in compositional turnover under climate change.

We found that the full set of concepts required to provide these explanations becomes available only at the end of our introductory section. We therefore provide the suggested explanations in a new paragraph preceding the Results section.

3. In the “A new methodology” subsection of the Results, also explain more clearly the relevance of the acceleration in GSAT increase after 1975 to the presented analysis of rates of species turnover before and after this time point given that (judging from Fig1) the former is an acceleration in a trend playing out over a much longer time scale than the latter. Given the 5-year time window over which species turnover was assessed appears to correspond more to the year-to-year fluctuations around the GSAT trend depicted in Fig 1, rather than the trend itself, a more astute reader is likely to wonder why the authors are placing so much emphasis on the change in long-term GSAT trend rather than the change (if any) in variability around this trend before-versus-after 1975. This issue deserves more attention.

We agree. A clearer explanation of our thinking regarding this point will help readers follow our argument. As we explained meticulously in our first response to Reviewer 3, the expected lags in the responses of populations and communities to changes in local climates will generally smooth out the effects of short-term fluctuations, exposing potential effects of long-term climatic trends.

Because of this tight connection between delay and smoothing of population responses, we integrated the suggested explanations into the first paragraph of Discussion, where we discuss delays and had previously already alluded to the associated smoothing.